# Diagnosis and Simultaneous Treatment of Musculoskeletal Injury Using H_2_O_2_-Triggered Echogenic Antioxidant Polymer Nanoparticles in a Rat Model of Contusion Injury

**DOI:** 10.3390/nano11102571

**Published:** 2021-09-30

**Authors:** Gi-Wook Kim, Nan-Hee Song, Mi-Ran Park, Tae-Eon Kim, Da-Sol Kim, Young-Bin Oh, Dong-Won Lee

**Affiliations:** 1Department of Physical Medicine and Rehabilitation, Jeonbuk National University Medical School, Jeonju 54097, Korea; murunoon@naver.com; 2Research Institute of Clinical Medicine of Jeonbuk National University—Biomedical Research Institute of Jeonbuk National University Hospital, Jeonju 54097, Korea; 3Department of Bionanotechnology and Bioconvergence Engineering, Jeonbuk National University, Jeonju 54896, Korea; nanhee9710@gmail.com (N.-H.S.); anne0189@naver.com (M.-R.P.); rotqhf94@jbnu.ac.kr (T.-E.K.); 4Department of Physical Medicine and Rehabilitation, Umji Clinic, Boryung 33522, Korea; youngbiin@hanmail.net; 5Department of Polymer Nano Science and Technology, Jeonbuk National University, Jeonju 54896, Korea

**Keywords:** musculoskeletal injury, vanillin, polymer nanoparticles, ultrasound imaging, inflammation

## Abstract

Ultrasound is clinically used for diagnosis and interventions for musculoskeletal injuries like muscle contusion, but contrast of ultrasonography still remains a challenge in the field of the musculoskeletal system. A level of hydrogen peroxide (H_2_O_2_) is known to be elevated during mechanical tissue damage and therefore H_2_O_2_ can be exploited as a diagnostic and therapeutic marker for mechanical injuries in the musculoskeletal system. We previously developed poly(vanillin-oxalate) (PVO) as an inflammation-responsive polymeric prodrug of vanillin, which is designed to rapidly respond to H_2_O_2_ and exert antioxidant and anti-inflammatory activities. The primary aim of this study is to verify whether PVO nanoparticles could serve as contrast agents as well as therapeutic agents for musculoskeletal injuries simultaneously. In a rat model of contusion-induced muscle injury, PVO nanoparticles generated CO_2_ bubbles to enhance the ultrasound contrast in the injury site. A single intramuscular injection of PVO nanoparticles also suppressed contusion-induced muscle damages by inhibiting the expression of pro-inflammatory cytokines and inflammatory cell infiltration. We, therefore, anticipate that PVO nanoparticles have great translational potential as not only ultrasound imaging agents but also therapeutic agents for the musculoskeletal disorders such as contusion.

## 1. Introduction

Closed muscle contusion or trauma injury is a relatively common in the musculoskeletal disorders. Acute tissue injuries occur when sudden and heavy forces are directly applied to the muscles. Chronic tissue injuries are caused by overuse, repetitive stress, sports activates, or incorrect mechanical movement [1,2,3]. The incidence varies from 10% to 55%, accounting for about 40% of musculoskeletal traumas in the young populations [1,3,4]. The clinical symptoms of contusion injuries are swelling, redness, local pain and tenderness during palpation. In severe cases, muscle weakness and decreased range of motion may occur [4].

The pathological progression following muscle contusion can be divided into three distinguished stages. First, the tissue destruction and inflammatory response occur 1 to 3 days after injury, and a large number of inflammatory cells infiltrate into the injury site. The second is the recovery phase (3 to 4 weeks), in which the regeneration of the disrupted myofibers and the formation of a connective tissue scar occur. The final is the remodeling stage (3–6 months), in which recovery of muscle function occurs through maturation of regenerated muscle fibers and reorganization of scar tissues [3,5,6]. In addition, inflammatory cells in injured tissues produce a large amount of reactive oxygen species (ROS), which contribute to chronic inflammation and also lead to delayed myalgia and muscular contraction. Among various ROS, hydrogen peroxide (H_2_O_2_) is considered the most abundant and plays important roles in inflammatory signaling [7]. In this regard, H_2_O_2_ in the inflamed tissues can be exploited as a diagnostic and therapeutic marker for the management of musculoskeletal injury.

Among several imaging modalities, ultrasound (US) and magnetic resonance imaging (MRI) are the main diagnosing tool for muscle contusion injuries. US using a high frequency transducer is basically easy to use and more cost effective than MRI. In addition, US can be performed with intervention simultaneously [1,7,8]. Several studies have recently reported that elastographic US can detect pathologic changes after muscle injury [7,9,10]. Gas-filled microbubbles have been commonly used as US contrast agents to enhance the contrast in vascularity and microcirculation of internal organs such as liver, pancreas, and kidney [11]. There are, however, little studies to report the clinical use of US contrast agents for the detection of musculoskeletal injuries.

Treatments of the muscle contusion injury include RICE (rest, ice, compression, elevation), exercise, physical therapy and therapeutics such as non-steroidal anti-inflammatory drug, corticosteroid and growth factor. As several studies suggested that musculoskeletal regeneration and healing are interfered with inflammation, anti-inflammatory therapeutics are commonly used to treat musculoskeletal injuries [12,13]. Recently, non-steroidal anti-inflammatory drugs have been widely used to relieve pain of acute or chronic musculoskeletal disorders with or without inflammation. However, long-term use of non-steroidal anti-inflammatory drugs has potential risk of gastrointestinal toxicity, hepatotoxicity and nephrotoxicity [13]. US-guided intervention in musculoskeletal disorders is safe and can accurately deliver drugs to the injured sites, thereby increasing the therapeutic effect. Corticosteroids such as dexamethasone and triamcinolone have been also used as an injectant. However, the routine use of corticosteroid is known to impair the skeletal muscle regeneration and also induce musculoskeletal adverse reactions including avascular necrosis of the bone, osteoporosis and tendinopathies [14,15,16]. Therefore, there is great need to develop a new therapeutic for musculoskeletal injuries.

Vanillin is one of major components of the extract of vanilla bean that has been extensively used as a flavoring agent in food, cosmetics, and pharmaceutical products [17,18]. Besides its various industrial applications, vanillin is also known to exert antioxidant, anti-inflammatory, antimicrobial and anticancer activities [19,20,21,22]. Vanillin has antioxidant activities by scavenging superoxide anion and hydroxyl radical and inhibiting protein oxidation and lipid peroxidation [18,23,24]. Vanillin also exhibits potent inhibitory effects on the expression of a number of inflammatory mediators and pro-inflammatory cytokines such as IL(interleukin)-1β, IL-6 and tumor necrosis factor-alpha (TNF-α) [23,25]. Interestingly, vanillin is also known to alter tissue blood flow to promote muscle relaxing effects through intramuscular vasodilation [26]. Like other natural compounds that inspired a number of US FDA-approved drugs, however, there are critical bottlenecks for using vanillin such as instability under physiological conditions. 

In order to overcome the limitations of vanillin and expand its therapeutic applications, we previously developed poly(vanillin-oxalate) (PVO) as an inflammation-responsive polymeric prodrug of vanillin that incorporates both H_2_O_2_-responsive peroxalate linkages and acid-cleavable acetal linkages in its backbone. PVO was designed to degrade by inflammatory stimuli such as acidic pH and H_2_O_2_ to release vanillin, leading to H_2_O_2_ elimination and anti-inflammatory actions. We demonstrated that PVO nanoparticles hold great potential as therapeutic agents for H_2_O_2_-associated pathological conditions such as acute liver failure and hepatic ischemia/reperfusion injury. PVO nanoparticles were also able to undergo H_2_O_2_-triggered oxidation to generate CO_2_ bubbles which enhance the US contrast in H_2_O_2_-rich hepatic injury.

Based on the notions that musculoskeletal injury is closely interrelated with inflammation which is characterized by high level of H_2_O_2_, we reasoned that H_2_O_2_-responsive PVO nanoparticles could serve as US contrast agents as well as therapeutic agents for H_2_O_2_-rich musculoskeletal disorders, simultaneously. In this study, the diagnostic and therapeutic activities of H_2_O_2_-triggered echogenic antioxidant PVO nanoparticles were evaluated using a rat model of contusion-induced muscle injury because a rat contusion model has been widely used to elucidate the pharmacological and physiotherapeutic procedures and is closely associated with skeletal muscle injuries [27]. In addition, the musculoskeletal anatomy and pathology are relatively similar to humans and the results in rat models could promote translation into clinical practices. NIH3T3 mouse fibroblasts were also used to evaluate the antioxidant and anti-inflammatory effects of PVO nanoparticles as fibroblasts comprise a port of cell populations in skeletal muscles and play important roles in maintaining muscle structure [28]. We constructed a null hypothesis that PVO nanoparticles are unable to elevate ultrasound contrast and exert therapeutic effects in a H_2_O_2_-triggered manner. The hypothesis was tested by measuring the ultrasound intensity and the levels of pro-inflammatory mediators and apoptosis-related genes of contusion-induced muscle injury. The results of the experimental studies suggest H_2_O_2_-triggered echogenic antioxidant PVO nanoparticles hold great translational potential as an ultrasonographic contrast agent and therapeutic agent for musculoskeletal injuries.

## 2. Methods

### 2.1. Preparation and Characterization of PVO Nanoparticles

PVO was synthesized from the reaction of oxalyl chloride, and acid-cleavable vanillin derivative as previously reported [23]. In brief, and acid-cleavable vanillin derivative (3.941 mmol) was placed in a flask containing 25 mL dried dichloromethane and pyridine (9.8 mmol). The flask was placed in ice bath and added with oxalyl chloride (3.941 mmol). The reaction was continued for 6 h at room temperature. PVO was obtained from extraction using dichloromethane/water and precipitated in cold hexane. After purification, the chemical structure of PVO was verified by NMR. PVO nanoparticles were prepared by a single emulsion method as previously reported [23]. Indocyanine green (ICG)-loaded PVO nanoparticles were also prepared for fluorescence imaging. Unloaded ICG was removed by centrifugation twice. The PVO nanoparticles suspended in water were characterized using a scanning electron microscopy (SUPRA40VP, Carl Zeiss, Jena, Germany) and particle size analyzer (Brookhaven Instruments, Holtsville, NY, USA). UV-vis absorption of the ICG-loaded PVO nanoparticles was investigated using a spectrometer (S-3100, Scinco, Seoul, Korea). 

### 2.2. Antioxidant and Anti-Inflammatory Activities of PVO Nanoparticles on Fibroblast Cells 

NIH3T3 cells were purchased from Korean Cell Line Bank. Cells were cultured in a plate containing Dulbecco’s Modified Eagle Medium (Gibco, Gland Island, NY, USA) containing 10% fetal bovine serum. In the presence of vanillin and PVO nanoparticles, cells were treated with 50 μM H_2_O_2_. At 24 h post-treatment, cell viability was determined by the conventional MTT (3-(4,5-dimethylthiazol-2-yl)-2,5-diphenyltetrazolium bromide) assay. The ROS level in cells were assessed using DCFH-DA (dichlorofluorescein-diacetate). 

### 2.3. A Rat Model of Skeletal Muscle Contusion Injury

Sprague Dawley rats (8-week-old males, Orient Bio, Seongnam, Korea) were anesthetized by intraperitoneal administration of the mixture of ketamine and xylazine (8:1). After anesthetization, isoflurane (Ifran Solution, Hana Pharm, Seoul, Korea) was delivered to rats with 100% oxygen using a vaporizer for maintenance of anesthetization. Rats were placed in the lateral decubitus position, with the knee extended and the ankle dorsiflexed. A compression instrument (TMS-PRO^®^, FTC Corp., Sterling, VA, USA) was used to induce contusion injury in triceps surae muscles. The muscles were pressed by a round flat probe (160-N force) descending at a rate of 20 mm/min until the thickness became 3 mm. Contusion injury was induced to the middle of the knee joint and the Achilles tendon. The injury site was marked by a Devon surgical skin marker for US imaging and biopsies. 

### 2.4. Fluorescence Imaging and US Imaging of Contusion Injury

ICG-loaded PVO nanoparticles were suspended in a saline solution at a concentration of 10 mg/mL. After anesthetization, the suspended nanoparticles (100 μL) were directly injected into the injury site at 1 day post-induction of contusion injury. The fluorescence images of the contusion injury were obtained using a fluorescent imaging system (FOBI, CELLGENTEK, Daejeon, Korea) at various different time points. For US imaging, PVO nanoparticles (150 μL of 5 mg/mL in saline) were directly injected into the peri-injury site 1 day after the induction of contusion injury. An US imaging instrument (Zone Ultra, Zonare Medical Systems, San Francisco, CA, USA) was used with a hockey stick style linear-array transducer (5 to 14 MHz). Dimension of the probe was 62 mm × 10 mm and the viewing width was 55 mm. The US images were obtained from long and short axis view by a physician with an extensive experience in musculoskeletal ultrasonography. All animal experiments were approved by the Institutional Animal Care and Use Committee (IACUC) of Jeonbuk National University Hospital (CUH-IACUC-2018-20-1) and conducted in accordance with the regulation and guideline.

### 2.5. RNA Isolation and Real-Time Reverse Transcription Polymerase Chain Reaction (RT-PCR)

After induction of contusion injury, PVO nanoparticles (25, 50 or 100 μL of 10 mg/mL) and vanillin (100 μL of 4 mg/mL) were directly injected into the injury site. At 3 days post-injection of PVO nanoparticles and equivalent vanillin (n = 6), the contusion induced triceps surae muscles were excised and homogenized using a TRIzol reagent (Life Technologies, Gaithersburg, MD, USA). Total RNA was extracted from the homogenized tissues using an RNA Iso kit (Takara, Kusatsu, Japan). After precipitation with isopropanol, RNA was dissolved in diethylpyrocarbonate-treated distilled water. Total RNA (2 μg) was treated with RNase-free DNase and first-strand cDNA was generated with oligo dT-adaptor primers by reverse transcription. Specific primers were designed using Primer3web (Table 1). Real time RT-PCR was performed using a mixture (10 μL) containing 10 ng of reverse-transcribed total RNA, 200 nM of forward and reverse primers and PCR master mixture. The mRNA levels of inducible nitric oxidase (iNOS), TNF-α, IL-1β, IL-6 and iNOS were quantitatively determined with glyceraldehyde-3-phosphate dehydrogenase (GAPDH) as a control. RT-PCR was carried out in 384-well plates using a real-time PCR system (ABI Prism 7900HT, Applied Biosystems, Foster City, CA, USA).

### 2.6. Western Blot Analysis

The muscle tissues were gently washed with an ice-cold phosphate buffer. The tissues were homogenized and treated with a lysis buffer (M-PER, Pierce Biotechnology, Rockford, IL, USA) for 20 min in an ice bath. The tissue lysates were centrifuged at 10,000× *g* and the obtained pellets were lysed with a RIPA lysis buffer (150 mM NaCl, 50 mM Tris-HCl pH 8.0, 1% NP-40, 0.5% sodium deoxycholate, 0.1% sodium dodecyl sulfate) containing protease and phosphatase inhibitors. After determining the concentration of proteins using a modified Bradford assay, 20 µg of protein were heat-treated for 5 min in loading buffer and separated by sodium dodecyl sulfate-polyacrylamide gel electrophoresis with 10% resolving gel and 3% acrylamide stacking gel. The separated proteins were transferred to polyvinylidene fluoride membrane. Membranes were blocked with TBS-T skim milk powder for 1 h at room temperature. The membrane was incubated with primary antibodies for Bax, caspase-3 (Cell Signaling, Danvers, MA, USA), Bcl-2, GAPDH (Bioworld, Bloomington, MN, USA), caspase-9, PI3K and Akt (Abcam, Cambridge, MA, USA) at 4 °C. Horseradish peroxidase-conjugated IgG (Zymed, South San Francisco, CA, USA) was used as a secondary antibody. The protein expression was visualized using a Luminescent image analyzer (LAS-1000, Fujifilm, Tokyo, Japan).

### 2.7. Histological Analysis

Muscle tissues were excised at 3 days after PVO nanoparticles injection. The tissues were fixed in a formaldehyde (4%) solution and embedded in a paraffin. The tissues were sliced 4 μm thick and the sections were mounted on the glass slide. The sectioned tissues were stained with hematoxylin and eosin (H&E), TUNEL (terminal deoxynucleotidyl transferase dUTP nick end labeling), Masson’s Trichrome and F4/80 antibody. 

### 2.8. Statistical Analysis

The statistical analyses were performed using SPSS (Version 18.0). Quantification in RT-PCR and western blot assay was performed using ImageJ and the one-way ANOVA with Scheffe method as post hoc test. The *p*-values less than 0.05 (95% confidence level) were considered statistically significant (*n* = 4).

## 3. Results

### 3.1. Characterization of PVO Nanoparticles

As shown in Figure 1a, PVO was synthesized from the polymerization reaction of oxalyl chloride and acid-cleavable vanillin derivative. Vanillin is incorporated in the backbone of PVO via an acid-cleavable acetal linkage and H_2_O_2_-responsive peroxalate ester linkage. PVO nanoparticles were monodispersed round spheres with a mean hydrodynamic diameter of ~330 nm (Figure 1b,c). Figure 1d illustrates the ability of PVO nanoparticles to eliminate H_2_O_2_. PVO nanoparticles eliminated H_2_O_2,_ concentration dependently (Figure 1d) because of H_2_O_2_-consuming oxidation of peroxalate esters.

To further investigate the H_2_O_2_-responsiveness of PVO nanoparticles, we also prepared autoquenched ICG-loaded PVO nanoparticles that contain a large amount of ICG. ICG-loaded PVO nanoparticles were also round nanospheres with a mean diameter of ~250 nm (Figure 2a,b). The loading of ICG in PVO nanoparticles was confirmed by the UV-vis spectrum and fluorescence imaging (Figure 2c). ICG-loaded PVO nanoparticles showed a broad absorption from 690 nm to 900 nm due to the presence of ICG. The absorption of ICG-loaded PVO nanoparticles was red shifted compared to free ICG because of physical interactions between PVO and fluorophore ICG. ICG-loaded PVO nanoparticles showed time- and H_2_O_2_-dependent fluorescence emission when excited with near infrared light (Figure 2d).

After 5 min incubation with H_2_O_2_, the fluorescence of ICG-loaded PVO nanoparticles increased because ICG sequestered in the hydrophobic interior was released to recover its fluorescence when PVO underwent H_2_O_2_-triggered hydrolytic degradation. In contrast, ICG-loaded PVO nanoparticles showed slightly increasing fluorescence with time in the absence of H_2_O_2_, indicating the H_2_O_2_-responsiveness of PVO nanoparticles.

### 3.2. Antioxidant and Anti-Inflammatory Activities of PVO Nanoparticles In Vitro

The antioxidant activity of PVO nanoparticles was evaluated using murine fibroblast NIH3T3 cells. Upon treatment with H_2_O_2_, the cell viability was dramatically reduced (Figure 3a). Vanillin showed marginal effects on the cell viability. In contrast, PVO nanoparticles effectively protected cells from H_2_O_2_-mediated toxicity. PVO nanoparticles at 100 μg/mL showed significantly stronger cytoprotective effects than the equivalent amount of vanillin (250 μM). H_2_O_2_-stimulated cells produced a large amount of ROS, evidenced by the strong green fluorescence of DCFH. While vanillin suppressed the ROS production to some extent, PVO nanoparticles at concentrations higher than 100 μg/mL suppressed ROS production almost completely (Figure 3b). The stronger cytoprotective effects of PVO nanoparticles over the equivalent amount of vanillin can be explained by the additive effects of H_2_O_2_-scavenging peroxalate and the intrinsic activity of vanillin.

### 3.3. US Imaging of PVO Nanoparticles in Agarose Gel Phantom

Figure 4 shows US images of PVO nanoparticles in the presence or absence of H_2_O_2_. In the absence of H_2_O_2_, PVO nanoparticles were not echogenic. In contrast, PVO nanoparticles showed a gradually increasing US contrast in the presence of 1 mM H_2_O_2_, with the highest echo signal at 30 min. With the higher concentration (5 mM) of H_2_O_2_, PVO nanoparticles displayed the higher echo signal, but the echo signal decayed faster. To further confirm H_2_O_2_-triggered echogenicity of PVO nanoparticles, PLGA, poly(lactic-co-glycolic acid) was used as a control, which does not respond to H_2_O_2_ nor generate CO_2_ bubbles. The same concentration of PLGA nanoparticles showed no echo signal in the presence of 5 mM H_2_O_2_. These findings demonstrate that the echogenicity of PVO nanoparticles is H_2_O_2_ concentration- and time-dependent.

### 3.4. Fluorescence Imaging of Contusion-Induced Triceps Surae Muscle Injury

It is known that skeletal muscle injury is known to generate a large amount of ROS including H_2_O_2_. In order to investigate whether PVO nanoparticles degrade in musculoskeletal disorders, we performed fluorescence imaging after direct injection of ICG-loaded PVO nanoparticles in the contusion-induced muscle injury. Contusion injury was induced on the left triceps surae muscle using a compression instrument (Figure 5a) and ICG-loaded PVO nanoparticles were directly injected into both triceps surae muscle at 1 day-post contusion injury. As expected from Figure 2c, we found that while the non-injured muscle showed no changes in the fluorescence intensity for 1 h, the injured muscle showed the gradually increasing fluorescence intensity (Figure 5b,c). The results suggest that PVO nanoparticles undergo H_2_O_2_-triggered degradation to release autoquenched ICG, which in turn recovers its fluorescence in the contusion-induced H_2_O_2_ rich muscle injury.

### 3.5. US Imaging of Contusion-Induced Triceps Surae Muscle Injury

H_2_O_2_-triggered echogenicity of PVO nanoparticles was assessed using a rat model of contusion injury (Figure 6a). US imaging was started as soon as PVO nanoparticles were injected directly around the muscle with contusion injury. Prior to the injection of PVO nanoparticles, the US imaging revealed a slightly hypoechoic area in the lesion of muscle tissues, but it was not prominent. No apparent echogenic signal was observed in the injured muscle. In contrast, significantly enhanced ultrasonographic contrast was observed immediately around the injected area after the injection of PVO nanoparticles. The echogenic PVO nanoparticles remained in the injured area at 5 min after the injection and they showed noticeable and lesion-specific contrast enhancement in the injured site. The follow-up ultrasonography displayed slow and gradual reduction of echo signal for 180 min (Figure 6b). When PLGA was injected, there is no enhancement in contrast (Figure 6c). To verify the H_2_O_2_-triggered echogenicity of PVO nanoparticles in the contusion-induced muscle, ultrasonography was made after injection of H_2_O_2_-scavenging catalase. Upon catalase injection, the echo signal of PVO nanoparticles rapidly diminished and no echo signal was observed at 15 min post-injection (Figure 6d). These results demonstrate that PVO nanoparticles become echogenic in a H_2_O_2_-triggered fashion. 

### 3.6. Therapeutic Effects of PVO Nanoparticles on Contusion-Induced Triceps Surae Muscle Injury

To evaluate the anti-inflammatory activities of PVO nanoparticles in the contusion injury of triceps surae muscles, we investigated the mRNA expression of pro-inflammatory cytokines by RT-PCR assay. Figure 7 shows the level of pro-inflammatory cytokines 3 days after the direction injection of vanillin or PVO nanoparticles (0.25, 0.5 and 1 mg) into muscles. Contusion injury in triceps surae muscles remarkably elevated the level of mRNA of iNOS, IL-1β, IL-6 and TNF-α, compared to the sham. However, single injection of PVO nanoparticles into the site of contusion injury significantly suppressed the expression of these pro-inflammatory cytokines, dose dependently. Vanillin (0.4 mg) also suppressed these cytokines, but less effective than equivalent PVO nanoparticles (1 mg).

We next performed the Western blot assay to examine the anti-apoptotic effects of PVO nanoparticles in the skeletal muscle injury. After the induction of contusion injury in triceps surae muscles, the expression of Bax, cleaved caspase-3 and caspase-9 was significantly upregulated, while the expression of Bcl-2, PI3K and Akt was downregulated, which is the typical phenotypes associated with apoptosis (Figure 8). However, PVO nanoparticles (1 mg) significantly suppressed contusion-mediated apoptosis in a dose dependent manner, evidenced by the downregulation of Bax, cleaved caspase-3 and caspase-9 and upregulation of Bcl-2, PI3K and Akt. Vanillin also exerted anti-apoptotic activity to some extent, but less effective than equivalent PVO nanoparticles.

### 3.7. Histological Examination of Contusion-Induced Triceps Surae Muscle Injury

Histological examination of muscle tissues was performed to further assess the therapeutic effects of PVO nanoparticles in contusion injury. Figure 9a shows the H&E-stained muscle tissues. Muscle with contusion injury showed severe damages to fibrillary texture of muscle along with massive infiltration of inflammatory cells. PVO nanoparticles remarkably suppressed the contusion-mediated muscle damages, with stronger therapeutic effects than the equivalent vanillin. Masson trichrome staining revealed that contusion injury induced fibrosis of muscle, evidenced by the transient increase of collagen deposition between myofibers [29,30]. Collagen deposition was effectively suppressed by PVO nanoparticles (Figure 9b). Infiltration of inflammatory cells into the lesion was confirmed by staining a macrophage marker, F4/80 antibody (Figure 9c). While vanillin showed moderate effects, PVO nanoparticles significantly suppressed the contusion-induced muscle damages and macrophage infiltration. Contusion injury also induced apoptotic cell death in muscles, evidenced by the massive TUNEL-positive cells (Figure 9d). PVO nanoparticles markedly suppressed the apoptotic cell death in the contusion-induced muscle.

## 4. Discussion

In the diagnosis of musculoskeletal disorders, US and MRI are commonly used clinically. Compared to MRI, US has the following advantages: a short period of examination time, high spatial resolution, being dynamic study, and low expenses. Above all, US is often used to guide various interventions immediately [31,32,33]. The musculoskeletal US can allow perilesional injection to the injured site at the same time as the diagnosis of the lesion. In the musculoskeletal disorders, US-guided intervention improves the therapeutic effect of drugs by allowing accurate injection into the lesion site compared to blind injection through physical examination [34,35,36]. However, despite patients’ apparent symptoms, no abnormal findings are observed in US imaging. In addition, it may be difficult to identify small lesions or damages such as repetitive micro-tear of the tendon or muscle in US imaging. In this case, small lesions can be diagnosed by confirming the patient’s response after US-guided injection into the clinically suspected lesion site [37,38].

The clinically used US contrast agents are microbubbles that have a diameter ranging from 1 to 8 μm. When these microbubbles are administered intravenously, they are confined to the blood pool and then can be easily detected by US imaging through strong interaction with the US beam. Commercially available microbubble contrast agents have been used clinically for cardiac pathology and imaging of the liver, breast, kidney, and urinary tract and the examples include SonoVue (sulphur hexafluoride microbubbles), Lumason (sulphur hexafluoride lipid-type A microspheres), Optison (perflutren protein-type A microspheres injectable suspension) and Definity (perflutren lipid microspheres) [11,39,40]. However, the use of ultrasound imaging based on blood pooling of microbubble contrast agents is limited in the detection of musculoskeletal disorders because musculoskeletal disorders are mainly focal lesions by damages. In the focal muscle or tendon injury, it is difficult to achieve the sufficient contrast effect using the microbubble ultrasound contrast agents because the blood supply to the focal damaged area is made by micro-vessels not by large blood vessels, unlike internal organs. Therefore, accurate diagnosis of musculoskeletal diseases requires ultrasonic contrast effects that are different from conventional gas-filled microbubbles and injection routes.

Although the pathophysiology of musculoskeletal disorders has not been clearly elucidated, the pathological progression following muscle contusion is divided into three stages: the tissue destruction and inflammatory response, the recovery phase, and the remodeling stage. Inflammation has been also assumed to be closely associated with musculoskeletal pain [13]. Inflammation is a dynamic process, responding to extrinsic/intrinsic damages and is also a major contributor to the pathogenesis of numerous musculoskeletal disorders [41]. Musculoskeletal disorders are primarily determined by inflammation and prolonged local musculoskeletal disorders are closely intertwined with chronic inflammation in the pathological progression to permanent damages and disability [42]. Currently, non-steroidal anti-inflammatory drugs have been widely used to control pain of acute or chronic musculoskeletal disorders with or without inflammation [12,13]. However, chronic use of non-steroidal anti-inflammatory drugs has potential risk of gastrointestinal toxicity, hepatotoxicity and nephrotoxicity [13]. Corticosteroids are also frequently prescribed to reduce pain and inflammation in the musculoskeletal systems because of their excellent anti-inflammatory activity. Injection is the most common method to deliver corticosteroids such as dexamethasone [16]. Injection is sometimes guided by US imaging to ensure safe and precise delivery of drugs and enhance the therapeutic efficacy. However, the routine use of corticosteroids is known to impair the skeletal muscle regeneration and also induces musculoskeletal adverse reactions including avascular necrosis of the bone, osteoporosis and tendinopathies [14,15]. Therefore, it is necessary to develop a new therapeutic agent alternative to the conventional injectants.

Naturally occurring bioactive compounds are a chemical substance produced by plants or microorganisms and have pharmacological and toxicological effects [43,44]. Despite great advance in synthetic chemistry as a tool to discover new drugs, naturally occurring compounds have been the successful source of drug candidates and their contribution to drug discovery and disease treatment is still enormous [45]. One of naturally occurring bioactive compounds that have recently drawn attention in biomedical applications is vanillin (4-hydroxy-3-methoxybenzylaldehyde), a phenolic aldehyde possessing a methoxy group and a hydroxyl group. As PVO was developed as an inflammation-responsive polymeric prodrug of vanillin and H_2_O_2_-triggered CO_2_ bubble generating polymer, in this study, we assessed the translational potential of PVO nanoparticles as an US contrast agent as well as a therapeutic agent of musculoskeletal disorders. 

In a rat model of contusion-induced muscle injury, perilesional injection of PVO nanoparticles increased echogenicity in the contusion area on US. The distinct echo signal was observed for more than 2 h and then gradually diminished. We previously reported that PVAX nanoparticles containing peroxalate esters in the polymeric backbone could elevate the ultrasound contrast for 6 h [7]. Although direct comparison was not made, PVO nanoparticles appeared to become echogenic faster and the echo signal also decayed faster than PVAX. This could be explained by the rationales that PVO has a higher content of aryl peroxalate than PVAX, which is responsible for H_2_O_2_-triggered degradation and CO_2_ bubble generation [46]. While PVO has 50% of aryl peroxalate, PVAX has 25% of aryl peroxalate. However, direct comparison studies are needed to evaluate their translational potential as a H_2_O_2_-responsive US contrast agent. Under the same condition, PVO nanoparticles could rapidly generate CO_2_ bubbles from H_2_O_2_-triggered oxidation of peroxalate esters and also rapidly lose their structural integrity. The echo signal of PVO nanoparticles in muscles lasted for 2 h, which is longer than those in the agarose gel phantom. The discrepancy in US imaging times can be explained by the different environments. While water in the well of agarose gel phantom is an open system, muscle tissue has a lower level of H_2_O_2_ and is a highly condensed closed system, which helps PVO nanoparticles hold CO_2_ bubbles for a longer period time. The pathological stimulus-triggered echogenicity and prolonged imaging time would be of great help to physicians in diagnosing small and mild musculoskeletal injuries. 

PVO nanoparticles exerted highly potent therapeutic activities in contusion injury by suppressing the expression of inflammatory cytokines such as TNF-α, IL-1β, IL-6 and iNOS and regulating apoptosis-related proteins. PVO nanoparticles also effectively inhibited the infiltration of inflammatory cells such as macrophage. Vanillin exerted anti-inflammatory effects to some extent, but less effective than PVO nanoparticles. The most appealing feature of PVO nanoparticles is their ability to scavenge H_2_O_2_, which acts as a second messenger in intracellular signal transduction pathways for pro-inflammatory mediators such as TNF-α, IL-1β and iNOS [47]. Therefore, the superior therapeutic activity of PVO nanoparticles over the equivalent vanillin is attributable to the additive effects of H_2_O_2_ scavenging and vanillin released in the pathological site. The sufficient therapeutic efficacy of single injection of PVO nanoparticles could be explained by the notions that PVO could scavenge pro-inflammatory H_2_O_2_ and release vanillin, as high as 30 wt% of its weight only in response to pathological stimuli. In this context, the null hypothesis was rejected by the statistically significant therapeutic outcomes and the elevated ultrasound contrast in a H_2_O_2_-rich musculoskeletal injury. It can be also reasoned that vanillin-releasing PVO nanoparticles have muscle relaxing effects as Peretti et al. reported that vanillin promotes muscle relaxing effects through intramuscular vasodilation [26]. However, further studies are warranted to elucidate the antiangiogenic and vasodilating potential of PVO nanoparticles in musculoskeletal disorders.

The translational potential of H_2_O_2_-triggered echogenic antioxidant PVO nanoparticles was assessed using a rat model of contusion-induced triceps surae muscle injury. Upon perilesional injection, PVO nanoparticles generated a significantly enhanced echo signal and alleviated contusion-induced muscle damages by suppressing the expression of pro-inflammatory cytokines such as TNF-α, IL-1β, IL-6 and iNOS and inhibiting apoptotic cell death. Although this study confirmed the potential of PVO nanoparticles as ultrasonographic contrast agents and therapeutics in a rat model of muscle injury, additional studies including long term toxicity and optimal doses are warranted to fully determine their translational potential. Large animal studies using a more clinically relevant model are also greatly needed prior to clinical translations.

## 5. Conclusions

This work is the beginning of translational studies to explore the applications of H_2_O_2_-triggered echogenic antioxidant PVO nanoparticles for the diagnosis and treatment of musculoskeletal disorders. PVO nanoparticles generated a significantly enhanced echo signal and the distinct echo signal lasted for 3 h. PVO as an inflammation-responsive polymeric prodrug of vanillin markedly suppressed the expression of pro-inflammatory cytokines and apoptotic cell death, with superior therapeutic effects than the equivalent amount of vanillin. Based on the H_2_O_2_-triggered echogenicity and excellent anti-inflammatory effects, PVO nanoparticles hold great potential as US contrast agents as well as therapeutic agents for various musculoskeletal disorders. The findings in this clinically relevant rat model demonstrate that H_2_O_2_-triggered echogenic and anti-inflammatory PVO nanoparticles could be translated into clinical applications.

## Figures and Tables

**Figure 1 nanomaterials-11-02571-f001:**
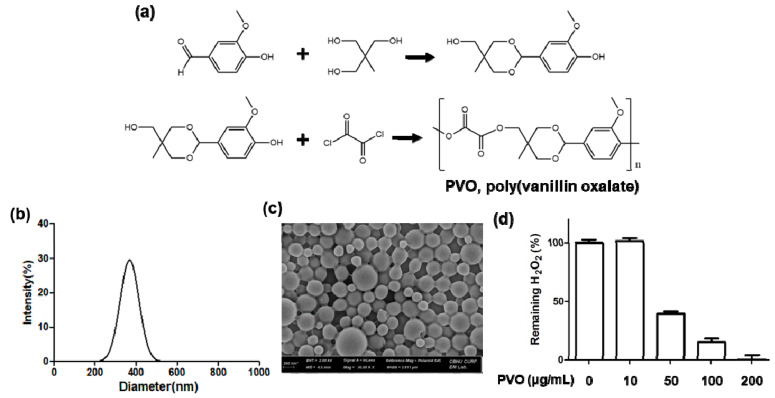
Characterization of PVO nanoparticles. (**a**) A synthetic route of PVO. (**b**) Hydrodynamic mean diameter and distribution of PVO nanoparticles. (**c**) SEM image of PVO nanoparticles. (**d**) Ability of PVO nanoparticles to scavenge H_2_O_2_.

**Figure 2 nanomaterials-11-02571-f002:**
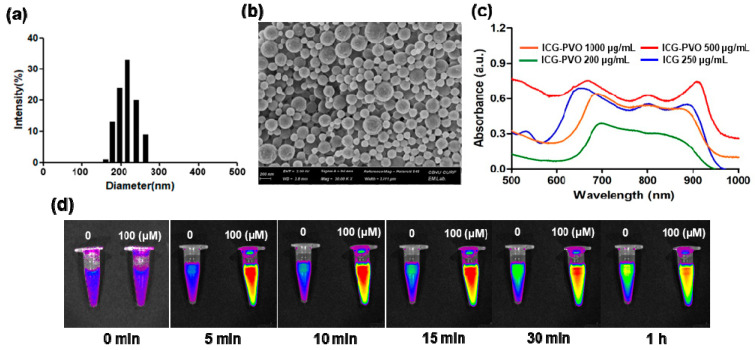
Characterization of ICG-loaded PVO nanoparticles. (**a**) Hydrodynamic mean diameter and distribution of ICG-loaded PVO nanoparticles. (**b**) SEM image of ICG-loaded PVO nanoparticles. (**c**) UV-vis absorbance of ICG-loaded PVO nanoparticles. (**d**) The effects of H_2_O_2_ on the fluorescence of autoquenched ICG-loaded PVO nanoparticles.

**Figure 3 nanomaterials-11-02571-f003:**
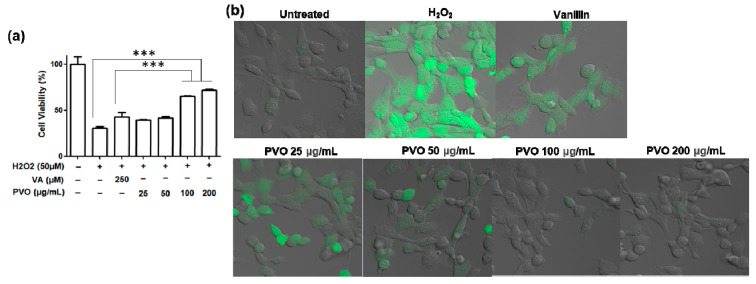
The effects of PVO nanoparticles on fibroblast cells. (**a**) Viability of NIH3T3 cells after the treatment of PVO nanoparticles. Values are mean ± s.d. (*n* = 4). *** *p* < 0.001. ANOVA was performed to determine the differences between groups using GraphPad Prism 8.0.1. (**b**) Suppression of ROS production in H_2_O_2_-stimulated cells.

**Figure 4 nanomaterials-11-02571-f004:**
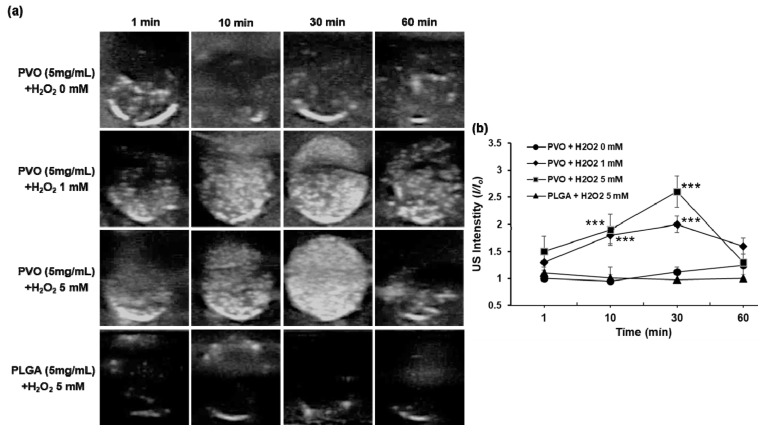
US images of PVO nanoparticles in agarose gel phantom. (**a**) Ultrasound images of CO_2_-bubble generating PVO nanoparticles. (**b**) Quantification of ultrasound intensity. Values are mean ± s.d. (*n* = 4). *** *p* < 0.001, relative to PVO + H_2_O_2_ 0 mM.

**Figure 5 nanomaterials-11-02571-f005:**
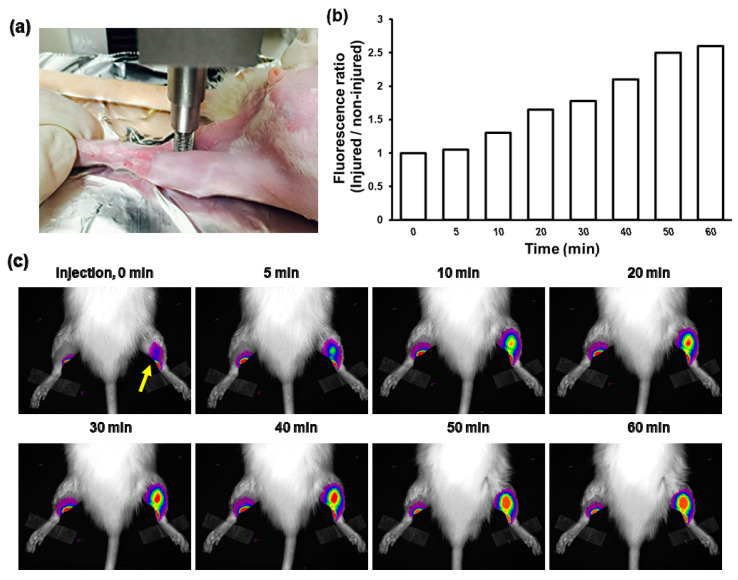
Detection of H_2_O_2_ in contusion-induced muscle injury. (**a**) Photograph showing the induction of contusion injury using a compression instrument. (**b**) Quantification of fluorescence intensity in the injured muscles. (**c**) Fluorescence images of contusion-induced triceps surae muscle injury at different time points. An arrow indicates the site of contusion-induced muscle injury.

**Figure 6 nanomaterials-11-02571-f006:**
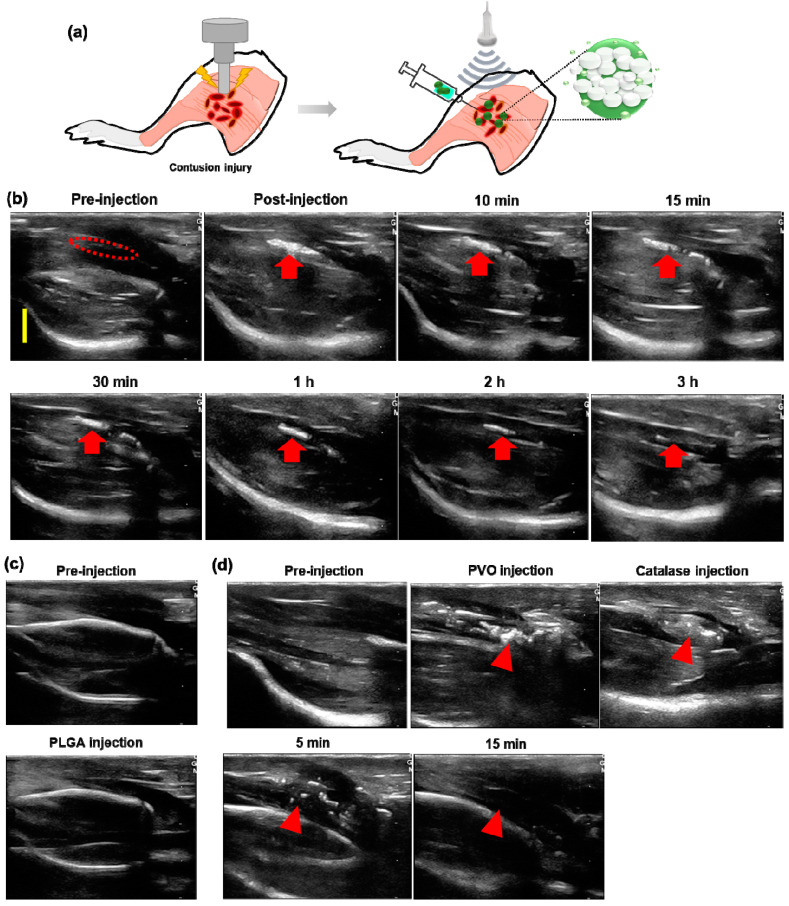
Ultrasonography of contusion-induced muscles. (**a**) Schematic illustration of contusion injury and CO_2_ bubble generating PVO nanoparticles as an ultrasound contrast agent. (**b**) Time course of US imaging at the contusion-induced muscles (red dotted line and arrows) after injection of PVO nanoparticles. The yellow scale bar is 5mm. (**c**) Ultrasonography of contusion-induced muscles before and after injection of PLGA nanoparticles. (**d**) Time course of US imaging at the contusion-induced muscles (red arrow-heads) after injection of PVO nanoparticles and catalase.

**Figure 7 nanomaterials-11-02571-f007:**
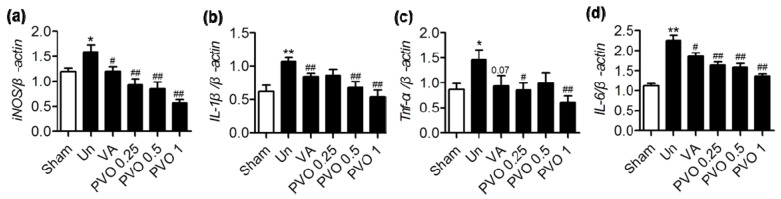
The mRNA levels of pro-inflammatory cytokines determined by RT-PCR at 3 days after treatment of PVO nanoparticles. (**a**) iNOS, (**b**) IL-1β, (**c**) TNF-α, (**d**) IL-6. Values are mean ± s.d. (*n* = 4). Values are mean ± s.d. (*n* = 4). Un and VA denotes untreated and vanillin, respectively. *** < 0.05 and ** *p* < 0.01 vs. Sham. # *p* < 0.05, ## *p* < 0.01 vs. Untreated.

**Figure 8 nanomaterials-11-02571-f008:**
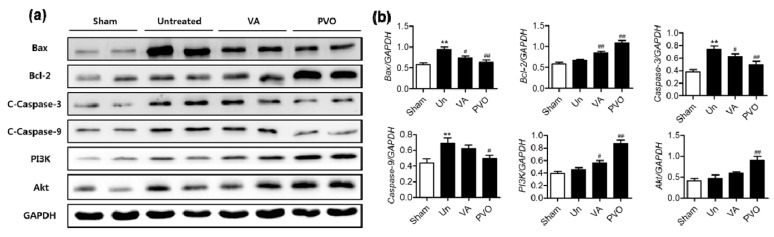
Therapeutic effects of PVO nanoparticles in the contusion-induced triceps surae muscle injury. (**a**) The levels of apoptosis-related genes. (**b**) Quantification of apoptosis-related genes. Un and VA denotes untreated and vanillin, respectively. ** *p* < 0.01 vs. Sham. # *p* < 0.05, ## *p* < 0.01 vs. Untreated.

**Figure 9 nanomaterials-11-02571-f009:**
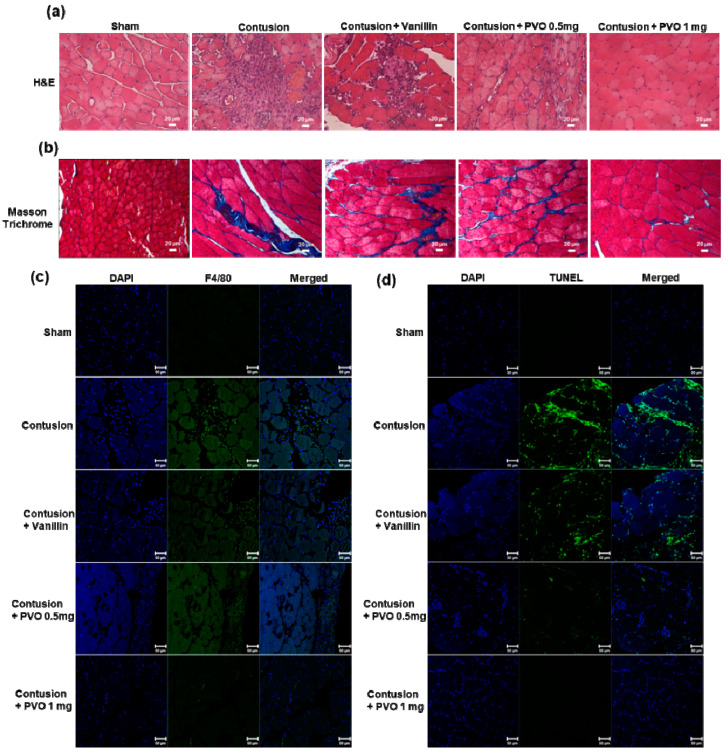
Histological examination of contusion-induced muscle. (**a**) H&E-stained tissues and (**b**) Masson trichrome- stained tissues. (**c**) F4/80-stained tissues. (**d**) TUNEL-stained tissues.

**Table 1 nanomaterials-11-02571-t001:** Sequences and accession numbers for primers (forward, FOR; reverse, REV) used in real-time RT-PCR.

Gene	Sequences for Primers	Accession No.
iNOS	FOR: AGGGAGTGTTGTTCCAGGTG	NM_012611
	REV: TCCTCAACCTGCTCCTCACT	
TNF-α	FOR: TGATCCGAGATGTGGAACTG	NM_012675
	REV: CCCATTTGGGAACTTCTCCT	
IL-1β	FOR: CAGGAAGGCAGTGTCACTCA	NM_031512
	REV: AGACAGCACGAGGCATTTTT	
IL-6	FOR: AGTTGCCTTCTTGGGACTGA	NM_012589
	REV:CAGAATTGCCATTGCACAAC	
Β-actin	FOR: TGTCACCAACTGGGACGATA	NM_031144
	REV: GGGGTGTTGAAGGTCTCAAA	

## Data Availability

The data used to support the findings of this study are available from the corresponding authors upon request.

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
