# Peer review of "Diagnosis and Simultaneous Treatment of Musculoskeletal Injury Using H2O2-Triggered Echogenic Antioxidant Polymer Nanoparticles in a Rat Model of Contusion Injury"

_nanomaterials, 2021, doi:10.3390/nano11102571_

Round 1
Reviewer 1 Report
TITLE: Diagnosis and simultaneous treatment of musculoskeletal injury using H2O2-triggered echogenic antioxidant polymer nanoparticles in a rat model of contusion injury
nanomaterials-1380345
The aim of the present investigation was to evaluate the effectiveness antioxidant polymer nanoparticles in a rat model of contusion injury
GENERAL COMMENTS
The article is in-line with the journal topic, but some minor methodologic flaws should be improved. The investigation is interesting and the results correctly described and commented in the discussion section. The present paper is recommended for publication to the present journal after minor revision.
Abstract
The section is a little be confused and should be rewritten according to the following subparagraph:
Introduction/aim of the study, materials and methods, results and conclusion.
Introduction
The rat model should be described in relation of the translational value of the experiment.
The null-hypothesis should be introduced at the end of the paragraph.
Materials and methods
Did you performed a sample size test to determine the specimens quantity?
No figures of the experiment on rats has been provided. The authors could improve the iconography.
I suppose that the application of an ANOVA test should be anticipated by a normality test (Shapiro-Wilks?).
Results.
A detailed descriptive statistic should be added (95% CI, range).
The SEM images should be improved in quality.
Fig.6 Why did you not performed a doppler technique for ultrasonography of contusion-induced muscles?
Discussion
The null-hypothesis should be discussed in this part of the manuscript.
Conclusions
The conclusion should improve the translational clinical application of your study.

Author Response
Abstract
- The aim of this study was included.
Introduction
- The translational value of the rat model was described.
- The null hypothesis was described.
Materials and methods
- The sample size test was not necessary because we performed animal studies with small sample size (n=4).
- A picture showing the induction of contusion injury was already included in Figure 5a. A schematic illustration was included to illustrate that PVO nanoparticles generate CO2 bubbles in H2O2-triggered manner in the muscle injury (Figure 6a).
- As addressed in 2.8 Statistical analysis, one-way ANOVA was performed with normality assumption because the animal study had small sample size.
Results.
- It was addressed in in 2.8 Statistical analysis. p values were used to determine the statistical significance (95% confidence level).
- We presented a new SEM image with better quality (Figure 1c).
- A Doppler ultrasound is used to image the blood flow by bouncing high-frequency acoustic wave off circulating red blood cells. Contusion injury in triceps surae muscle is not related with blood flow. We therefore did not perform Doppler ultrasound imaging.
Discussion
- We included a statement regarding the null-hypothesis.
Conclusions
- We included a statement saying that the results in clinically relevant animal study demonstrate then great translational potential in clinical applications.
Reviewer 2 Report
The paper is well written, the topic is very interesting and The study represents the starting point for translational studies to explore PVO Nanoparticles as US contrast agents and therapeutic agents for the musculoskeletal disorders. A lot of experiments have been done and results well discussed . therefore, the paper deserves to be published.
Author Response
No request for revision
This manuscript is a resubmission of an earlier submission. The following is a list of the peer review reports and author responses from that submission.
Round 1
Reviewer 1 Report
The introduction is well structured and the aim of the work is clearly described. A lot of experimental work has been done; however, the section with the data of the materials used is missing, and concerning section 2.1, about the preparation of the nanoparticles, there is confusion between what has already been published and what seems new in this manuscript. Particularly, authors should describe ICG-PVO nanoparticles preparation and characterization if they prepare and characterize them for the first time here (or at least include satisfactory references).
It is extremely important to demonstrate that ICG is loaded into nanoparticles, in which amount and that fluorescence detected in diverse experiments is due to ICG-nanoparticles and not to ICG released from nanoparticles.
Authors should also determine the size and size distribution of ICG-nanoparticles as well as morphological properties by SEM or TEM
Different properties of fluorescent nanoparticles compared to PVO nanoparticles could cause different behavior of the nanosized systems.
I suggest correcting Figure 1as follows:
Figure 1. Preparation and characterization of PVO nanoparticles. (a) A synthetic route of PVO. (b) The hydrodynamic mean diameter and size distribution of PVO nanoparticles. (c) SEM image of PVO nanoparticles (magnification……add the value). (d) The ability of PVO nanoparticles to scavenge H2O2.
Figures 1e and 1 f should be moved to a new section concerning ICG-PVO nanoparticles preparation and characterization
Reference needs to be revised: journal abbreviations do not conform to the style required from the journal, and mistakes in authors’ names, lack of the page range are just some examples.
Reviewer 2 Report
TITLE: Polymeric Microspheres/Cells/Extracellular Matrix Constructs Produced by Auto-Assembly for Bone Bodular Tissue Engineering
The aim of the present investigation was to investigate the diagnostic and therapeutic activities of H2O2-triggered echogenic antioxidant PVO nanoparticles were evaluated using a rat model of contusion-induced muscle injury.
GENERAL COMMENTS
The article is in-line with the journal topic, but some minor methodologic flaws should be improved. The investigation is interesting and the results correctly described and commented in the discussion section. The present paper is recommended for publication to the present journal after major revision.
Title: The title should include the kind of investigation that the authors performed (systematic review, animal study, clinical trial….)
Introduction
The authors should discuss the null-hypothesis of the present study and the medical implications of the theorem that they want to introduce.
The rat model and the NIH3T3 cells lineage should be justified according the translational model for a human application.
Materials and methods
Did you performed a sample size test to determine the specimens quantity?
How did you obtained the NIH3T3 cells?
“A compression instrument (TMS-PRO® 134 , FTC Corp., Sterling, VA, USA) was used to induce contusion in135 jury in triceps surae muscles. The muscles were pressed by a round flat probe (160-N force) 136 descending at a rate of 20 mm/min until the thickness became 3 mm”. The authors should provide detailed iconography and figures of the surgery
The biopsies timepoints are missed in 2.3 paragraph. Did the animals euthanized?
Results
In fig. 1c the authors provided a SEM images and spectroscopy, but they did not described adequately this part in materials and methods.
The Fig. 5 is too small and low in contrast to evaluate the US findings.
Discussion
The null-hypothesis should be discussed in this part of the manuscript.
The limitations of the present study should be discussed in this section.
Conclusions
The conclusion should reflect a translational clinical application of your study.
